# PREFERENCE-ENHANCED INSTRUCTION TUNING FOR MACHINE TRANSLATION

## ABSTRACT

Although Large Language Models (LLMs) like GPT-4 perform excellently in machine translation, their high costs and scalability make them unavailable in many scenarios. Recently, there has been increased effort to build smaller LLMs that can achieve comparable performance. However, while typical instruction tuning methods tend to directly mimic reference translations, leading to less meaningful results, recent preference optimization methods have shown improvements. Despite this, they still fail to effectively utilize crucial preference information during inference. In this paper, we introduce **P**reference-**E**nhanced **I**nstruction **T**uning (PEIT), a novel method that explicitly incorporates preferences into both the instruction fine-tuning and the inference phase. Our extensive experiments show that PEIT not only improves translation quality but also significantly outperforms state-of-the-art preference optimization methods and instruction tuning baselines on multiple language benchmarks.

## 1 INTRODUCTION

Large language models (LLMs), such as GPT-4 (Achiam et al., 2023), have been showing predominant performance in machine translation (MT) (Hendy et al., 2023; Zhu et al., 2023; Jiao et al., 2023b). However, attaining such level of performance often requires the expense of substantial model size, significant infrastructure demands, and high deployment costs. To address these challenges, recent research has ***shifted toward fine-tuning smaller LLMs*** to enhance translation capabilities while mitigating the associated resource overhead. (Zeng et al., 2023; Jiao et al., 2023a; Kudugunta et al., 2024; Zan et al., 2024; Li et al., 2024; Guo et al., 2024; He et al., 2024; Wu et al., 2024a; Xu et al., 2024b). For example, ALMA (Xu et al., 2023) enhances the multilingual capabilities of LLaMA-2 (Touvron et al., 2023b) by fine-tuning with non-English data and refining it with high-quality translation instruction data. Similarly, Aya (Aryabumi et al., 2024) fine-tunes smaller LLMs using a larger amount of translation instruction examples from the Aya Dataset (Singh et al., 2024), allowing it to achieve stronger translation performance.

Simple instruction tuning using translation pairs has its limitations, primarily due to the quality issues inherent in the reference data — even when it is human-generated (Xu et al., 2024b; He et al., 2024; Wu et al., 2024b). These imperfections can impede the LLM's ability to produce high-quality translations, as it may merely learn to replicate the references during instruction tuning. To address this limitation, recent works have moved beyond direct instruction tuning, focusing instead on preference optimization (Zhu et al., 2024; He et al., 2024; Xu et al., 2024c; Wang & Xiao, 2024; Xu et al., 2024b). For example, Contrastive Preference Optimization (CPO) (Xu et al., 2024b) is one of the leading approaches, which enables the LLM to learn from preferences between synthesized preference-rich translation pairs, allowing it to exceed the quality of the original reference data by optimizing based on comparative judgments rather than simple replication. In human translation, context is often leveraged to enhance translation accuracy(House, 2006). Intuitively, machine translation should benefit from contextual information in a similar way. However, even with preference optimization methods, this information is not effectively utilized due to the issue of prompt shift (Li et al., 2023), which occurs when the preference intentions (e.g., contextual information or translation examples) embedded in the inference prompt misalign with the model's training data. This misalignment makes it difficult for the model to capture and incorporate these intentions, leading to translation bias and outputs that deviate from the expected behavior.

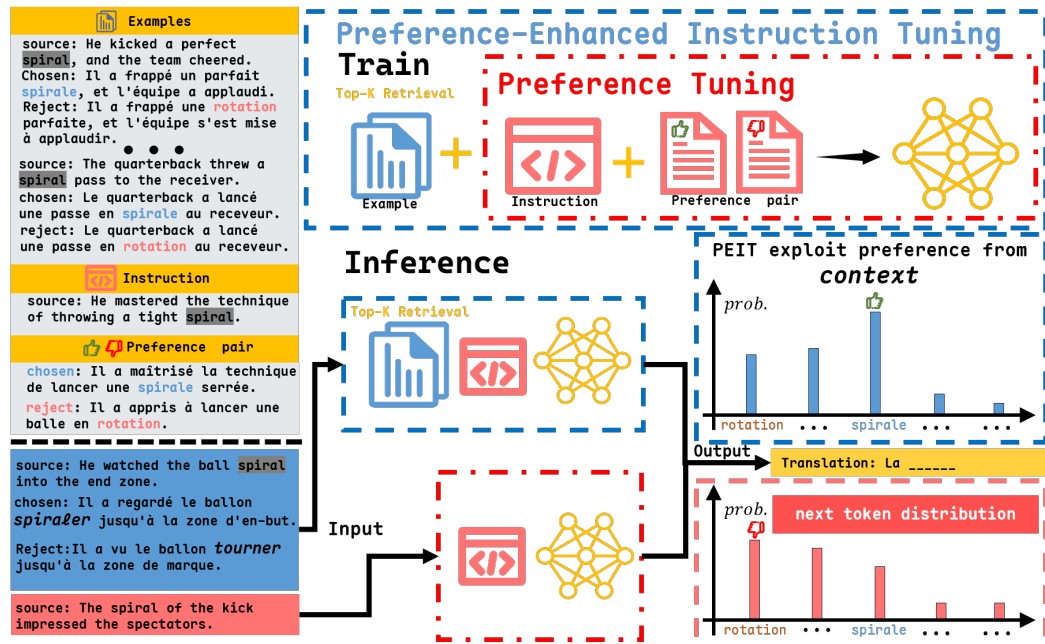

Figure 1: We compare **P**reference-**E**nhanced **I**nstruction **T**uning (PEIT) with other methods using blue and red . The ones in the upper left corner with yellow and gray backgrounds are the legend and legend examples. PEIT enables the translation model to learn how to catch preference intentions from the preference scenarios provided by the context during the training process. In the inference phase, it can easily identify the context to which the source belongs and generate translations that align with the preferences of the current scenario.

As shown in Figure 1, to overcome above issues, we propose using different strategies during the training and inference phase to catch intentions effectively. In the training phase, we initially re-trieve preference-rich translation pairs from the corpus that are contextually similar to the source text. These preference-rich translation pairs are then integrated with the source text to construct the model's training data, and then we propose the **P**reference-**E**nhanced **I**nstruction **T**uning (PEIT) technique to enable the model to naturally develop the ability to understand preferences in con-text. During the inference phase, we explicitly provide the retrieved preference-rich translation pairs along with the current source text as input, allowing the model to generate translations with the correct preference intention.

Our main contributions are summarized as follows:

- We introduce **P**reference-**E**nhanced **I**nstruction **T**uning(PEIT), which helps the model learn preference from retrieved preference-rich translation pairs, boosting the performance of preference alignment of the translation model.

- We have theoretically validated that PEIT can guide the model toward the preference sce-narios associated with the current source text from the perspective of parameter editing, enabling a single model to better adapt to translation tasks with different preference inten-tions.

- We conducted extensive comparative experiments to demonstrate that in preference data, reject items, like the chosen items, also significantly affect translation model performance.

## 2 PREFERENCE-ENHANCED INSTRUCTION TUNING

In this section, we first introduce the challenges of using a single model to handle complex transla-tion scenarios. Next, we discuss how, from a parameter update perspective, a customized prefix in

the model input can potentially mitigate these challenges. Finally, we present the modeling objective of PEIT, which focuses on learning preferences from the context to address these limitations.

## 2.1 THE PARADOX OF FITTING

Translation demands arise in diverse scenarios (e.g., written and spoken language) in different languages. Consequently, the data in translation tasks naturally exhibit inconsistent distributions, leading to varying preferences of dynamic scenarios. However, current post-training methods for adapting smaller LLMs to translation tasks involve only a single parameter edit (from $\theta$ to $\theta + \Delta\theta$). According to NFL theory (Wolpert & Macready, 1997), such models cannot perform optimally across all preference distributions. We prove (Appendix A.2 for more details) that there exists a set of model parameters $\{\theta + \Delta\theta_i\}$ such that the average loss measured across all distributions is strictly greater than the average loss measured when fitting each distribution with these independent models, differing by at least an insurmountable constant $\epsilon$ :

$$\frac{1}{k}\sum_{i=1}^{k}\mathcal{L}(f_{\theta+\Delta\theta}, D_i) > \frac{1}{k}\sum_{i=1}^{k}\mathcal{L}(f_{\theta+\Delta\theta_i}, D_i) + \epsilon \tag{1}$$

where, $D_i$ represents a preference distribution in a specific direction within the overall dataset $\mathcal{D}$, and $\mathcal{L}$ measures the error of the model $f$ on $\mathcal{D}$. In this light, although we aim to minimize loss across all datasets during the training process, it inevitably results in an **unbridgeable** lower bound on loss across all datasets.

Whereas, we can address this issue by introducing a mapping $g : \mathbb{R}^{|\theta|} \to \mathbb{R}^{|\theta|}$ in the parameter dimension, which offers a targeted parameter transformation when this single model computes on $D_i$ such that $g(\theta + \Delta\theta, D_i) = \delta\theta$ (where $\delta\theta$ represents a value in $\mathbb{R}^{|\theta|}$). This transformation allows the single model to achieve the same loss lower bound as a multi-model setup:

$$\min_{\theta}\frac{1}{k}\sum_{i=1}^{k}\mathcal{L}(f_{\theta+\Delta\theta+g(\theta+\Delta\theta,D_i)}, D_i) \iff \min_{\theta}\frac{1}{k}\sum_{i=1}^{k}\mathcal{L}(f_{\theta+\Delta\theta_i}, D_i) \tag{2}$$

## 2.2 CONTEXT PROVIDE FINE-GRAINED FINE-TUNING GRADIENT DIRECTION

We can formally prove that the translation model can learn a mapping $g$ from the context $C$, allowing the single model to better adapt to various preferences by using In-Context Learning (ICL) during inference. The main idea of the proof is to decompose the ICL mechanism into a direction-learnable parameter edit (Dai et al., 2023), which helps the smaller LLM make fine-grained tuning during computation, thereby aligning with different preference distributions.

Specifically, as $q = W_Q[C; x]$ is the vector of the attention query which constructed by source text $x$ and related Context $C$. We simulated the model's computational process after incorporating $C$, and derive from equation 3.

$$\mathcal{F}_{ICL}(\theta; [C; x]) = Attn(V, K, q) = W_V[C; x]softmax(\frac{(W_K[C; x])^T q}{\sqrt{d}}) \tag{3}$$

For ease of understanding, we analyze the approximation of standard attention by removing the scaling factor $\sqrt{d}$ and unary operations, transforming it into relaxed linear attention.

$$\mathcal{F}_{ICL}(\theta; [C; x]) \approx W_V[C; x](W_K[C; x])^T q := \hat{\mathcal{F}_{ICL}}(\theta; [C; x]) \tag{4}$$

We define $W_\theta = W_V x(W_K x)^T$ as the initialized parameters to be updated since $W_\theta q$ is the attention result in the zero-shot learning (ZSL) setting, where no $C$ is provided. Proceeding with the derivation:

$$\begin{aligned}
\hat{\mathcal{F}_{ICL}}(\theta; [C; x]) &= W_V[C; x](W_K[C; x])^T q \\
&= W_V x x^T W_K^T q + W_V C C^T W_K^T q \\
&= W_\theta q + W_V C C^T W_K^T q \Leftrightarrow W_\theta q + \Delta W q \\
&= (W_{\theta+\Delta\theta})q
\end{aligned} \tag{5}$$

This shows that we enable more fine-grained parameter editing, aligning the final effective parameters more closely with the desired distribution of preference by customizing the context $C$. Therefore, we can leverage the flexibility demonstrated by ICL to guide the translation model toward the preference distribution corresponding to the current source text.

### 2.3 Learning to Learn Preference from In-Context

Our primary goal is to guide the model to learn preference information from the CONTEXT, meaning that the model can learn a correct mapping $g$ from the preference-rich examples. We achieve this goal by training the model using a preference-enhanced ICL loss which contain a $\mathcal{L}_{prefer}$ and $\mathcal{L}_{context}$ to strengthen the model's ability to recognize preferences from the output perspective and robustness in utilizing explicit preference information.

Given a set of source sentences $x$, alongside preferred translation output $y_w$ and sub-preferred translation $y_l$, we can access a dataset, denoted as $\mathcal{D} = \bigcup_k D_k = \left\{ \left( x^{(i)}, y_w^{(i)}, y_l^{(i)} \right) \right\}_{i=1}^{N}$. We obtain samples from $\mathcal{D}$ that belong to the same preference distribution $D_i$ as the current source text $x_i$ to serve as the context $C_i$. Then, We minimize the objective $\mathcal{L}(f_{\theta+\Delta\theta+g(\theta+\Delta\theta,D_i)}, D_i)$ by optimizing the loss function:

$$\mathcal{L}_{ICL} = -\mathbb{E}_{(x,y_w,y_l)\sim\mathcal{D}}[log\pi_\theta(y_w|C,I,x)]$$

We use a preference loss $\mathcal{L}_{prefer}$ to enhance the model's ability to distinguish preferences in its output. This loss effectively approximated to DPO loss, helps PEIT learn the preferred translation and reject suboptimal translations, which can be defined as follows:

$$\mathcal{L}_{prefer} = -\mathbb{E}_{(x,y_w,y_l)\sim\mathcal{D}}[\log \sigma \left( \beta \log \pi_\theta(y_w|C,I,x) - \beta \log \pi_\theta(y_l|C,I,x) \right)]$$

Furthermore, PEIT's ability to identify intended preferences from retrieved examples relies heavily on the quality of these examples. Therefore, enhancing the model's robustness in handling low-quality examples is a critical issue that must be addressed. We have designed a training objective $\mathcal{L}_{context}$ that encourages the model to align the preference intention representations from examples of varying quality.

Let $h_C^i$, $h_{C^+}^i$, and $h_{C^-}^i$ denote the representations of the preferences intentions of the model for contextual information $C$, $C^+$, and $C^-$, respectively, which, despite differences in quality, share similar preference intention. To align the representation $h_C^i$ with $h_{C^+}^i$ and $h_{C^-}^i$, we optimize the model using a contrastive loss $\mathcal{L}_{context}^i$ defined as:

$$\mathcal{L}_{context}^i = -log \frac{e^{sim(h_C^i, h_{C^+}^i)/\tau}}{e^{sim(h_C^i, h_{C^+}^i)/\tau} + e^{sim(h_C^i, h_{C^-}^i)/\tau}},$$

where $sim(h_i, h_j)$ is the similarity of two preference representation, and $\tau$ is a temperature hyperparameter.

Combining the above all parts, the overall learning objective is

$$\min_\theta \mathcal{L}_{ICL} + \mathcal{L}_{prefer} + min(\lambda, \frac{\mathcal{L}_{ICL}}{\mathcal{L}_{context}})\mathcal{L}_{context}$$

where $\lambda$ controls the weight assigned to the context loss, balancing the model's ability to recognize preference intentions with overall translation quality. By aligning the hidden representations of contexts with similar preference tendencies, regardless of quality differences, the model becomes more robust in discerning desired preferences from low-quality examples.

## 3 Experiments

### 3.1 Preference data

We conduct main experiments on the ALMA-R-Preference dataset which (Xu et al., 2024a) released, and selected both the chosen and rejected translations for the target language based on the average quality of each data item. To demonstrate the generality of our approach, we also performed

supplementary experiments on translation tasks involving other low-resource languages using the Flores-200 dataset (Team et al., 2022). We transformed the Flores-200 dataset into a pairwise preference dataset by implementing a **synthetic preference data** method tailored for our experiments. Building upon the approach of (Nvidia et al., 2024), we adopted the **LLM-as-generator** method, utilizing different large language models to generate candidate responses. According to the definition in (Jiang et al., 2024), we employed a **Feedback from Inductive Biases** method to construct the preference direction. This ensures that the preference direction aligns to the test set provided in the Flores-200 dataset.

## 3.2 SETTINGS

We view PEIT not just as an optimization technique, but as a conceptual approach that highlights the importance of providing preference information during both the training and inference phases. This aids in aligning model outputs with desired translation tendencies and enhances overall performance. To demonstrate the potential of in-context preference learning, we developed a series of progressively comparative methods. Following this, we introduce the baselines chosen for our experiments and explain the rationale behind these selections.

**SFT** Using supervised fine-tuning (SFT) to adapt large language models to specific downstream tasks is a fundamental approach. Its effectiveness has been validated through extensive practical experiments. Therefore, SFT on prefer data serves as the first baseline in our experiments.

**CPO and DPO** We also compared the commonly used preference alignment methods in the machine translation field. These two methods are derived from the same optimization goal (Schulman et al., 2017) but reflect different training objectives due to the adoption of distinct assumptions. Therefore, these two methods serve as the primary comparative methods for preference alignment evaluation.

**ICFT and ICPFT** We set In-Context Fine-Tuning (ICFT) and In-Context-Preference Fine-Tuning (ICPFT) as baselines to compare with SFT and PEIT, demonstrating the validity of our approach and the superiority of PEIT. Specifically, in the SFT scenario, we add a retrieved example to each input during training to create the ICFT setup. For ICPFT, we enhance the example by incorporating preference pair.

**PE-CPO** Preference-Enhanced Contrastive Preference Optimization (PE-CPO) aims to ensure a fairer comparison and to strengthen the baseline, we introduced the concept of PEIT into the CPO method, resulting in the PE-CPO baseline. This enhanced version integrates preference estimation into the CPO framework, allowing the model to utilize in-context preference information during training, providing a more strong baseline against which to evaluate our proposed approach.

## 3.3 OTHER DETAILS

**Base model** Our experiment primarily focuses on comparing fine-tuning methods rather than specific base models. We conducted our main experiments on widely used open source LLMs(Touvron et al., 2023a; Dubey et al., 2024). To avoid data leakage, we used an earlier version, LLaMA2-13B, for our experiments and present the main result in it. We also conduct additional experiments with other models, and release result in Appendix E.

**Training with PEFT** During the training phase, we focus exclusively on updating the weights of the added LoRA parameters. These weights have a rank of 32 and only add an additional 24M parameters to the original 13B size of the model. The fine-tuning process involves a batch size of 32, spanning 5 epochs, and accommodating sequences with a maximum length of 512 tokens.

**Translation Instruction** Like the base model, the translation prompt is not our main focus, so it will not be carefully tuned. We will maintain consistency across all experiments, and details can be found in the Appendix C.

**Hyperparameters** For CPO, DPO and PEIT, we adhere to the default $\beta$ value of 0.1 as used by (Xu et al., 2024a; Rafailov et al., 2024). For PEIT, we set $\tau$ and $\lambda$ to 0.3 and 2.0. For all baselines, we set the learning rate to 2e-5. To ensure the fairness of the experiment, we have made a specific design for the DPO training process. Before applying the DPO method, we conducted preliminary training with the selected data to simulate the typical pipeline for preference alignment using DPO. We have provided more detailed settings in Appendix B.

Table 1: The main result in Translating from English (en→xx). PEIT methods significantly outperform all comparable methods. The dark blue boxes in ICFT and PECPO indicates a significant improvement compared to their original versions (ICFT and CPO), while light blue boxes represents only a small but noticeable enhancement. For PEIT, Dark blue boxes signifies a significant improvement compared to the second-best method in all comparisons, and light blue boxes follows the same pattern. All red colors indicate a slight decrease in performance.

| Methods | de | | zh | | ru | | cs | | ind | |
|---|---|---|---|---|---|---|---|---|---|---|
| | BLEU | XCOMET | BLEU | XCOMET | BLEU | XCOMET | BLEU | XCOMET | BLEU | XCOMET |
| SFT | 30.25 | 88.81 | 27.94 | 87.94 | 27.12 | 88.37 | 26.22 | 87.62 | 25.93 | 89.48 |
| DPO | 29.50 | 90.03 | 27.33 | 88.23 | 26.32 | 87.03 | 26.25 | 88.68 | 25.41 | 89.43 |
| CPO | 30.54 | 90.16 | 24.87 | 89.85 | 25.14 | 89.63 | 27.13 | 88.73 | 27.21 | 90.04 |
| ICFT | 29.19 | 89.24 | 24.77 | 88.12 | 24.54 | 88.78 | 25.06 | 86.64 | 24.95 | 89.33 |
| ICPFT | 29.96 | 89.91 | 25.78 | 88.37 | 27.93 | 89.13 | 28.25 | 87.28 | 25.37 | 89.21 |
| PE-CPO | 31.41 | 90.66 | 25.89 | 90.23 | 26.20 | 90.11 | 27.88 | 89.27 | 23.73 | 89.26 |
| PEIT | 31.22 | 91.74 | 26.33 | 90.51 | 27.22 | 90.47 | 29.47 | 89.53 | 26.01 | 90.13 |

**PEIT instantiation** Within the PEIT framework, various implementation options exist for each component. Here, we outline the specific details used in our experiments. In the implementation of the retriever, we use "xlm-r-bert-base-nli-stsb-mean-tokens" (Reimers & Gurevych, 2019) as the sentence embedding model and train a Faiss (Douze et al., 2024) index for similarity retrieval. When calculating $L_{context}$, we take the probability distribution of the model's first output token as $h_C$. We choose the cosine similarity function as the instantiation of $sim()$ to measure the similarity of $h_C$. For each text to be translated, we set the number of examples $k = 1$ during training, and we also report the results for the same $k$ during testing.

## 3.4 RESULTS

We present the primary result in Table 1 and Table 2, average score in Table 3. Our evaluation metrics include both statistical and neural metrics (Papineni et al., 2002; Rei et al., 2020), but we place a primary emphasis on neural metrics, using statistical metrics only as a reference with a limited level of confidence. For neural metrics, we adopted the XCOMET series models [1], and for statistical metrics, we used BLEU.

**Compared with implicit tuning methods** We first compared the results of PEIT with those of fitting-based fine-tuning methods. Under the evaluation of robust neural metrics, PEIT, due to its ability to leverage fine-grained preference information, achieved a higher average score compared to the average scores of CPO and DPO. In translation tasks in five languages, including German, Chinese, Russian, Czech, and Indonesian, PEIT achieved an average score of 92.10 of XCOMET, CPO averaged 90.92, DPO 89.43, and SFT 89.51. However, from our motivation's perspective, achieving an excellent average score does not necessarily mean the method is sufficient. Therefore, we conducted a more fine-grained analysis of the models' capabilities trained by different methods at Section 4. Overall, PEIT demonstrates a clear advantage over previous methods in fine-grained comparisons.

**Compared with ablation** We also conducted ablation experiments on our own design, and this section of the experiment demonstrated the effectiveness of our approach. After incorporating preference information into the task demonstration context (ICFT to ICPFT), we observed an improvement of 0.68% in our main evaluation metrics. Following the inclusion of our $\mathcal{L}_{context}$ design in PE-CPO, the score increased further by 0. 58%.

## 4 ANALYSES

The analysis in this section is divided into two parts. In the first part, we focus on evaluating the effectiveness of PEIT, first comparing the degree of improvement that PEIT brings to the model versus CPO and PE-CPO, and then conducting a performance analysis of the models trained with PEIT.

---

[1] we use XCOMET-XL

Table 2: The main result in Translating to English (xx→en).

| Methods | de | | zh | | ru | | cs | | ind | |
|---|---|---|---|---|---|---|---|---|---|---|
| | BLEU | XCOMET | BLEU | XCOMET | BLEU | XCOMET | BLEU | XCOMET | BLEU | XCOMET |
| SFT | 33.12 | 93.67 | 25.13 | 90.45 | 39.12 | 90.42 | 41.22 | 86.54 | 31.05 | 91.86 |
| DPO | 31.99 | 93.24 | 25.17 | 89.94 | 39.11 | 89.16 | 42.15 | 86.70 | 31.58 | 91.92 |
| CPO | 32.74 | 94.72 | 26.32 | 91.73 | 38.26 | 91.85 | 43.13 | 89.91 | 30.27 | 92.60 |
| ICFT | 31.45 | 93.57 | 24.78 | 90.84 | 33.14 | 91.28 | 39.57 | 87.82 | 33.51 | 93.73 |
| ICPFT | 31.19 | 95.43 | 25.78 | 91.76 | 35.54 | 91.97 | 40.13 | 88.72 | 35.02 | 94.39 |
| PE-CPO | 35.33 | 95.96 | 25.89 | 93.13 | 37.55 | 92.35 | 43.34 | 90.67 | 30.11 | 93.67 |
| PEIT | 34.21 | 96.31 | 26.22 | 92.86 | 39.13 | 93.44 | 41.47 | 91.37 | 37.47 | 94.71 |

Table 3: average result of the main experiment across two translation directions.

| XCOMET | SFT | DPO | CPO | ICFT | ICPFT | PE-CPO | PEIT |
|---|---|---|---|---|---|---|---|
| Translating from English (en→xx) | 88.44 | 88.67 | 89.68 | 88.42 | 88.77 | 89.90 | 90.47 |
| Translating to English (xx→en) | 90.58 | 90.19 | 92.16 | 91.44 | 92.45 | 93.15 | 93.73 |

In the second part, we provide a detailed description of our two methods for generating synthetic preference data and analyze the impact of synthetic preference data on the translation task.

### 4.1 DOES PEIT RECOGNIZE FINE-GRAINED PREFERENCE INTENTIONS?

We provide a comprehensive view of the impact of PEIT by examining fine-grained performance and deeper token distribution patterns. We adopt a more detailed comparative approach, going beyond average scores, to analyze the results of PEIT alongside other fine-tuning methods. The primary focus is the win rate ratio between PEIT and the alternative methods. By inferring preference tendencies from finely retrieved contextual information, PEIT demonstrates an ability to align with the current text more effectively. This allows it to outperform fitting-based fine-tuning methods.

**Ties-K win-rate curve** Specifically, we compare each score of generative translation, and to mitigate the inherent errors of the metric, we arrange the score differences in ascending order, designate the smallest k of differences as ties, and subsequently recalculate the win rate in Fig. 2. This approach allows us to plot a graph illustrating the relationship between k and the win rate. Judging the fine-grained win rate of two methods from the trend of the Ties-K win-rate curve is more fair and robust than directly comparing win rates.

By incorporating the contextual loss term $\mathcal{L}_{context}$ into the learning objective, the model can robustly leverage explicit preference information during inference phase. To evaluate this, we conducted experiments on a well-trained model by introducing controlled perturbations to the context and analyzing performance trends. This allows us to assess the model's sensitivity to changes in the contextual scene, which is critical for understanding how well the model adapts to subtle shifts in preference signals.

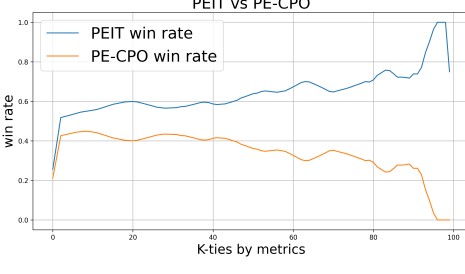

(a) train CPO with Preference Enhance  (b) train CPO without Preference Enhance

Figure 2: Ties-K win-rate curve of PEIT with CPO and PE-CPO. This curve indicates that the superiority of PEIT's average results is not driven by a few very high-score translations, but rather that nearly every translation achieves relatively high quality.

**Different preferences representations of context** To investigate how PEIT learns preference from context at varying levels, we visualized the hidden states of the final output tokens from the last Transformer layer of the decoder in Fig. 3. We selected 100 data points from the ALMA-R-Preference dataset (Xu et al., 2024a) and chose similar samples at different levels of contextual similarity, representing different distances in preference space. By capturing the hidden states from the final layer for each of these levels, we aim to understand how preference representation changes as the model encounters different contextual clues.

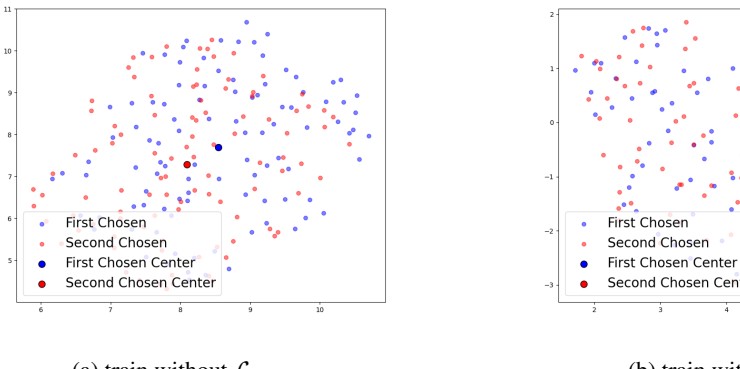

(a) train without $\mathcal{L}_{context}$            (b) train with $\mathcal{L}_{context}$

Figure 3: UMAP Visualization (McInnes et al., 2020) illustrates the preference intentions learned by the model trained with PEIT, with or without $\mathcal{L}_{context}$, based on preference examples of varying retrieval quality. The proximity of the two centers reflects the model's ability to effectively learn preferences from these examples. In this experiment, we represent the model's observed preference tendencies through the probability distribution of its first output token after reading the preference examples.

To provide a more intuitive understanding of the results, we applied dimensionality reduction to the token distributions, projecting the high-dimensional vocabulary space onto a lower-dimensional space suitable for visual analysis. This visualization allowed us to examine the model's output patterns under various contextual conditions.

In addition, we conducted an ablation experiment on in-context examples to investigate the impact of different qualities of in-context examples on the performance of PEIT. We concluded that the more complete the retrieval (using higher quality relevant examples or increasing the number of relevant examples), the better the final result. Detailed results can be found in Appendix D.

From this analysis, it becomes clear that PEIT not only effectively utilizes contextual preference information during inference phase but also demonstrates robustness to the quality of the context. Even when the preference information is weak or ambiguous, PEIT can still accurately discern and align with the desired preferences. This indicates that PEIT is capable of capturing fine-grained preference intentions, maintaining its performance even when contextual cues are less informative.

### 4.2 How preference data quality affects PEIT?

In this section, we explore how the quality of preference data affects the performance of PEIT. To investigate this, we employed two methods to generate preference data for our experiments. Using the **Feedback from Inductive Biases** perspective (Jiang et al., 2024), we transformed this generated data into preference datasets, allowing us to assess how variations in data quality influence PEIT's ability to learn and generalize preference intentions. From this perspective, we treat the original instruction data as embodying test-time preferences, and any generated translation—regardless of quality—as misaligned with these preferences due to differing inductive biases in the models. This approach allows us to construct preference datasets where the original instructions serve as the preferred option, while translations generated by both methods are considered less preferred.

Table 4: different quality preference data

| XCOMET | SFT | DPO | CPO | ICFT | ICPFT | PE-CPO | PEIT |
|---|---|---|---|---|---|---|---|
| GPT-Generated | 90.67 | 90.67 | 91.32 | 91.53 | 91.80 | 91.46 | 92.15 |
| Self-Paraphrasing | 90.67 | 88.93 | 90.66 | 91.53 | 91.12 | 90.65 | 91.37 |

**GPT-Generated Translations** The first method involves using GPT-4 to generate translations. We provided GPT-4 with the original instruction data, which represents the test-time preferences, and prompted it to produce translations. Although GPT-4 can generate high-quality, fluent, and grammatically correct translations, these outputs may not fully reflect the specific preferences encoded in the original instruction data due to differing inductive biases. We regard these GPT-generated translations as less preferred compared to the original instructions, as they may not capture the subtle preferences embedded in the instruction data.

**Self-Paraphrasing** The second method involves training the model on the test set and repeatedly generating translations through a paraphrasing process. Specifically, we fine-tuned the model on the training set using Supervised Fine-Tuning (SFT) to perform the translation task. By generating multiple translations of the same training set, we collected a series of paraphrased outputs. These paraphrased translations are syntactically different but semantically similar to the originals. However, since they are produced by the model itself, they may reflect the model's own inductive biases rather than the specific preferences encoded in the original instruction data. As a result, we treat these translated versions as less preferred options in our preference dataset.

We compared the results of all the baselines mentioned in this study on the preference datasets constructed using these two methods. The experiments demonstrate that higher-quality distractor data leads to higher-quality preference data, with GPT-4 generated data achieving better results across all methods. We specifically examine the differences between data generated using GPT-Generated and Self-Paraphrasing methods. The data generated by GPT shows superior quality in various dimensions compared to Self-Paraphrasing, while semantically it does not exhibit significant deficiencies compared to the original instruction data, differing only in preference tendencies.

Compared to CPO's approach of constructing preference data, where the lowest-quality data is used as the distractor in preference pairs, our experimental results seem to suggest a different recommendation. However, the underlying principle is the same: maximizing the quality of distractor data is essential to increasing the value of preference data.

## 5 CONCLUSION

In this paper,we introduce preference-enhanced instruction tuning (PEIT) as a novel approach to enhancing machine translation quality by leveraging in-context preference learning with large language models. Experiments across multiple languages demonstrate that PEIT outperforms existing methods like CPO, DPO, and SFT, especially when high-quality preference data is utilized. PEIT achieves such outstanding performance even without fully exploring the form of explicit preference information. Objectively speaking, PEIT is orthogonal to methods like CoT (Wei et al., 2023), ReFT (Luong et al., 2024), and even LLM reasoning. Overall, PEIT offers a promising direction for improving machine translation systems by effectively integrating explicit preference information through in-context learning.

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

## A PROOFS

### A.1 QUESTION

Let $D_1, D_2, \ldots, D_k$ be $k$ distinct data sets, each from a different preference distribution. We define the following two scenarios:

**Single Model:** We fit $f_\theta$ on all data sets using a single neural network to $f_{\theta+\Delta\theta}$.

**Multiple Models:** We fit $f_\theta$ on each data set $D_i$ using a separate neural network to $f_{\theta+\Delta\theta_i}$.

Let $\mathcal{L}(f_\theta, D_i)$ denote the loss function (e.g., croos-entropy loss) for fitting data set $D_i$ with the neural network $f_\theta$. The total loss for the single model is given by:

$$\mathcal{L}_{\text{total}} = \frac{1}{k} \sum_{i=1}^{k} \mathcal{L}(f_{\theta+\Delta\theta}, D_i), \tag{6}$$

and the loss for the multiple models scenario is:

$$\mathcal{L}_{\text{sep}} = \frac{1}{k}\sum_{i=1}^{k}\mathcal{L}(f_{\theta+\Delta\theta_i}, D_i). \tag{7}$$

Our goal is to prove the following two results:

1. $\mathcal{L}_{\text{total}} > \mathcal{L}_{\text{sep}} + \epsilon$, i.e., the error using a single model is higher than the average error using multiple models.

2. If we allow the parameters $\theta$ of the single model to be dynamically adjusted for each data set by a mapping, then the error lower bound for the single model can match that of multiple models.

## A.2 PROOF OF HIGHER ERROR IN SINGLE MODEL SCENARIO

We proof this by NFL theory:

$$Proof. \quad w.l.o.g. \quad \forall f_{\theta+\Delta\theta}, \ \exists f_{\theta+\Delta\theta_k} \quad s.t.\ \mathcal{L}(f_{\theta+\Delta\theta}, D_k) < \mathcal{L}(f_{\theta+\Delta\theta_k}, D_k) \quad (k \neq j) \tag{8}$$

Assemble a set of all $f$ that meet the conditions.

$$Let \quad F_k = \{f_\theta \mid \mathcal{L}(f_{\theta+\Delta\theta}, D_k) < \mathcal{L}(f_\theta, D_k)\} \tag{9}$$

From NFL theory, we know that:

$$\exists f_j \in F_k \quad s.t.\ \mathcal{L}(f_{\theta+\Delta\theta}, D_j) \geq \mathcal{L}(f_j, D_j) \tag{10}$$

Let $f_{\theta+\Delta\theta_j} = f_j$, we have:

$$\mathcal{L}(f_{\theta+\Delta\theta}, D_j) \geq \mathcal{L}(f_{\theta+\Delta\theta_j}, D_j) \tag{11}$$

By a similar argument, for each $i \in \{1, 2, ..., k\}$, we can find such a $f_i$ ,so that:

$$\mathcal{L}(f_{\theta+\Delta\theta}, D_i) \geq \mathcal{L}(f_{\theta+\Delta\theta_i}, D_i) \tag{12}$$

The equal sign is not held at the same time, so we have:

$$\frac{1}{k}\sum_{i=1}^{k}\mathcal{L}(f_{\theta+\Delta\theta}, D_i) > \frac{1}{k}\sum_{i=1}^{k}\mathcal{L}(f_{\theta+\Delta\theta_i}, D_i) + \epsilon \tag{13}$$

Thus, we conclude that using multiple models results in a lower average error than using a single model.

## A.3 PARAMETER ADJUSTMENT ADDRESSES GAP

Now, consider the case where the parameters $\theta$ of the single model can be dynamically adjusted for each data set. Let $\delta\theta_i = g(\theta + \Delta\theta, D_i)$, where $g$ is a function that generates the parameters $\theta_i$ for data set $D_i$, and $\phi$ represents shared global parameters.

The optimization problem is now:

$$\min_{\Delta\theta}\frac{1}{k}\sum_{i=1}^{k}\mathcal{L}(f_{\theta+\Delta\theta+g(\theta+\Delta\theta, D_i)}, D_i). \tag{14}$$

Let $g(\theta + \Delta\theta, D_i) = \Delta\theta_i - \Delta\theta$ , it can generate parameters $\theta + \Delta\theta$ to $\theta + \Delta\theta_i$ that are close to the optimal parameters for each data set. In that case, we have:

$$\nabla_\theta\mathcal{L}(f_{\theta+\Delta\theta+g(\theta+\Delta\theta, D_i)}, D_i) = \nabla_\theta\mathcal{L}(f_{\theta+\Delta\theta_i}, D_i),$$

which implies that the parameter adjustment allows the single model to achieve the same gradient updates as the multiple models. Therefore, the error lower bound of the single model with dynamic parameter adjustment is:

$$\min_\theta\frac{1}{k}\sum_{i=1}^{k}\mathcal{L}(f_{\theta+\Delta\theta+g(\theta+\Delta\theta, D_i)}, D_i) \iff \min_\theta\frac{1}{k}\sum_{i=1}^{k}\mathcal{L}(f_{\theta+\Delta\theta_i}, D_i) \tag{15}$$

# B    Experiments details of baseline

We provide a detailed explanation of our experimental design and procedure here. Our experiment aims to compare the effects of using preference data with different methods.

**SFT** We use the chosen entries in the pair-wise preference dataset as the labels for SFT, setting the learning rate to $2 \times 10^{-5}$, LoRA rank to 32, and LoRA alpha to 64. These hyperparameters are consistent with all other baselines.

**DPO** We use the chosen and reject entries in the pair-wise preference dataset as the labels for DPO's SFT init, setting the learning rate to $2 \times 10^{-5}$, LoRA rank to 32, and LoRA alpha to 64. Subsequently, we continue training the model with DPO, setting the reference for DPO as a duplicate of this SFT model.

**CPO** Since CPO inherently includes an SFT loss, we do not initialize it with SFT. Instead, we proceed directly with CPO training.

**ICFT, ICPFT and PE-CPO** We use the same retrieval method as PEIT, except that in ICFT, samples with preference information are replaced by those without preferences. Implementation details can be found in Appendix C.

Table 5: Baseline configurations and hyperparameters

| Baseline | lr | Lora rank | Lora target | Initialization | Random seed |
|----------|-----|-----------|-------------|----------------|-------------|
| SFT | 2e-5 | 32 | QKVO | Gaussian distribution | 42 |
| CPO | 2e-5 | 32 | QKVO | Gaussian distribution | 42 |
| DPO | 2e-5 | 32 | QKVO | Adapter weights trained with SFT | 42 |
| ICFT | 2e-5 | 32 | QKVO | Gaussian distribution | 42 |
| ICPFT | 2e-5 | 32 | QKVO | Gaussian distribution | 42 |
| PECPO | 2e-5 | 32 | QKVO | Gaussian distribution | 42 |
| PEIT | 2e-5 | 32 | QKVO | Gaussian distribution | 42 |

As a result, our controlled variable experiment limits most of the variable factors, which makes conclusions almost dependent on the training method.

# C    Translation Prompt

We designed different prompts, as shown in Fig. 4 for each method based on their characteristics and the required information, but we did not deliberately perform prompt engineering for each method in the experiments.

# D    Ablation of in-context example

We present the experimental setup, as shown in table 6, and results of our ablation experiment on in-context examples. In our experiments, we set k = 1 by default. Additionally, we evaluated the impact of different k values on performance, as shown in the table above. The larger the k value, the better the performance. However, since larger k values result in higher training and inference costs, we chose k = 1 as the default.

Table 6: Base Model Details

| Base model | Dataset | Direction |
|------------|---------|-----------|
| Llama3-8b | ALMA-R-Preference | xx→en |

Subsequently, we explored the impact of selecting examples with different similarity ranks on the performance of PEIT when k=1.

```
### Instructitn:
Translate the following input text into <target-language>.
### input:
<source-text>
### Output:
```
(Instruction Prompt — SFT, DPO, CPO)

```
Refer to the examples to complete following translation task.
### [Example1]
### Example input: <example source-text>
### Example output:<example target-text>
### Instructitn:
Translate the following input text into <target-language>.
### input:
<source-text>
### Output:
```
(In-Context Prompt — ICFT)

```
Refer to the examples to complete following translation task.
### [Example1]
### Example input: <example source-text>
### Example chosen:<example chosen-text>
### Example reject:<example reject-text>
### Instructitn:
Refer to the above examples to translate the text into
<target-language> and reduce the probability of rejection.
### input:
<source-text>
### Output:
```
(In-Context-Preference Prompt — PEIT, ICPFT, PE-CPO)

Figure 4: Prompt used in experiment

Table 7: PEIT performance for different values of $k$

| PEIT | k=1 | k=2 | k=3 |
|------|-----|-----|-----|
| XCOMET | 95.25 | 95.29 | 95.36 |

Table 8: PEIT performance for different rank of example

| Model | PEIT+rank 1 example | PEIT+rank 2 example | PEIT+rank 3example |
|-------|---------------------|---------------------|--------------------|
| XCOMET | 95.25 | 94.67 | 94.59 |

| Model | PEIT+constant example | SFT | CPO |
|-------|-----------------------|-----|-----|
| XCOMET | 93.60 | 92.13 | 93.62 |

As can be seen from the ablation experiments, as shown in table 7 and 8, the more complete the retrieval (using higher quality relevant examples or increasing the number of relevant examples), the better the final result.

## E  FULL EXPERIMENT RESULT

We present all our experimental results here, though the comparisons are not entirely fair, as they involve some additional models. Some results are from CPO, and we have also included Aya-23-8B, an advanced instruction-tuned model, whose test results show that it has reached GPT-4's performance level.

Table 9: Llama3-8B main result

| Methods | de BLEU | de XCOMET | zh BLEU | zh XCOMET | ru BLEU | ru XCOMET | cs BLEU | cs XCOMET | ind BLEU | ind XCOMET |
|---|---|---|---|---|---|---|---|---|---|---|
| *Translating from English (en→xx)* | | | | | | | | | | |
| SFT | 31.25 | 92.31 | 26.79 | 89.44 | 27.87 | 89.07 | 26.65 | 89.04 | 26.98 | 89.56 |
| DPO | 29.77 | 90.03 | 26.23 | 89.07 | 26.29 | 88.98 | 25.92 | 88.68 | 26.42 | 89.32 |
| CPO | 31.59 | 91.77 | 26.81 | 90.16 | 25.41 | 90.23 | 27.79 | 89.79 | 27.20 | 90.52 |
| ICFT | 30.09 | 89.45 | 25.88 | 88.73 | 23.21 | 89.39 | 26.36 | 89.15 | 23.99 | 89.59 |
| ICPFT | 30.78 | 90.23 | 26.20 | 89.24 | 26.53 | 89.92 | 27.31 | 89.82 | 26.34 | 89.83 |
| PE-CPO | 31.11 | 91.96 | 25.52 | 90.29 | 27.33 | 90.64 | 28.13 | 90.30 | 24.75 | 90.74 |
| PEIT | 31.24 | 92.63 | 27.16 | 91.01 | 28.10 | 91.13 | 29.34 | 91.21 | 27.22 | 91.21 |
| *Translating to English (xx→en)* | | | | | | | | | | |
| SFT | 35.65 | 94.97 | 26.42 | 91.45 | 40.26 | 92.17 | 43.97 | 87.62 | 37.62 | 93.41 |
| DPO | 33.90 | 94.74 | 24.26 | 90.93 | 37.38 | 91.06 | 40.12 | 86.70 | 32.73 | 92.92 |
| CPO | 35.61 | 95.72 | 25.39 | 92.69 | 38.04 | 92.72 | 43.18 | 88.21 | 37.31 | 94.03 |
| ICFT | 32.23 | 95.17 | 25.10 | 91.82 | 33.69 | 92.38 | 39.55 | 87.80 | 34.25 | 93.98 |
| ICPFT | 33.65 | 95.73 | 26.44 | 92.13 | 38.35 | 92.35 | 42.21 | 87.93 | 35.74 | 94.21 |
| PE-CPO | 36.43 | 96.39 | 26.86 | 93.41 | 38.92 | 93.29 | 42.35 | 90.80 | 36.21 | 94.58 |
| PEIT | 36.76 | 96.87 | 27.91 | 93.64 | 39.99 | 94.23 | 43.35 | 90.93 | 38.32 | 95.22 |

Table 10: The full result in xx → en including both statistic, reference-free and reference-based metrics.

| Models | de BLEU | de KIWI | de XCOMET | zh BLEU | zh KIWI | zh XCOMET | ru BLEU | ru KIWI | ru XCOMET |
|---|---|---|---|---|---|---|---|---|---|
| GPT-4 | 32.41 | 81.50 | 94.47 | 23.82 | 79.33 | 92.06 | 41.09 | 81.57 | 90.95 |
| Aya23-8B | 44.84 | 84.42 | 96.56 | 39.41 | 83.16 | 93.42 | 45.68 | 85.02 | 95.15 |
| LLaMA-2-13B | 31.06 | 79.47 | 91.10 | 21.81 | 75.09 | 85.68 | 36.50 | 79.14 | 86.12 |
| SFT on prefer | 33.12 | 84.01 | 93.67 | 25.13 | 81.53 | 90.45 | 39.12 | 81.92 | 90.42 |
| DPO | 31.99 | 82.91 | 93.24 | 25.17 | 81.73 | 89.94 | 39.11 | 81.40 | 89.16 |
| CPO | 32.74 | 83.70 | 94.72 | 26.32 | 82.37 | 91.73 | 38.26 | 82.58 | 91.85 |
| ICFT | 31.45 | 81.31 | 93.57 | 24.78 | 80.26 | 90.84 | 33.14 | 81.27 | 91.28 |
| ICPFT | 31.19 | 81.97 | 95.43 | 25.78 | 81.63 | 91.76 | 35.54 | 82.06 | 91.97 |
| PECPO | 35.33 | 82.21 | 95.96 | 25.89 | 81.31 | 93.13 | 37.55 | 82.31 | 92.35 |
| PEIT | 34.21 | 83.42 | 96.31 | 26.22 | 82.04 | 92.86 | 39.13 | 83.22 | 93.44 |

| Models | cs BLEU | cs KIWI | cs XCOMET | ind BLEU | ind KIWI | ind XCOMET | Avg. BLEU | Avg. KIWI | Avg. XCOMET |
|---|---|---|---|---|---|---|---|---|---|
| GPT-4 | 46.86 | 82.52 | 88.48 | 38.25 | 82.73 | 94.98 | 36.48 | 81.53 | 92.18 |
| Aya23-8B | 50.87 | 83.27 | 93.74 | 40.23 | 84.33 | 96.88 | **44.20** | **84.04** | **95.15** |
| LLaMA-2-13B | 40.02 | 79.29 | 78.50 | 30.03 | 73.20 | 88.72 | 31.88 | 77.23 | 86.02 |
| SFT on prefer | 41.22 | 80.93 | 86.54 | 31.05 | 81.51 | 91.86 | 33.92 | 81.98 | 90.58 |
| DPO | 42.15 | 80.26 | 86.70 | 31.58 | 81.85 | 91.92 | 34.00 | 81.63 | 90.19 |
| CPO | 43.13 | 81.74 | 89.91 | 30.27 | 82.36 | 92.60 | 34.14 | 82.55 | 92.16 |
| ICFT | 39.57 | 80.63 | 87.82 | 33.51 | 81.47 | 93.73 | 32.48 | 80.98 | 91.44 |
| ICPFT | 40.13 | 80.58 | 88.72 | 35.02 | 80.63 | 94.39 | 33.53 | 81.37 | 92.45 |
| PECPO | 43.34 | 81.52 | 90.67 | 30.11 | 82.52 | 93.67 | 34.44 | 81.97 | 93.15 |
| PEIT | 41.47 | 81.35 | 91.37 | 37.47 | 82.87 | 94.71 | **35.70** | **82.58** | **93.73** |

Table 11: The full result in en → xx including both statistic, reference-free and reference-based metrics.

| Models | de BLEU | KIWI | XCOMET | zh BLEU | KIWI | XCOMET | ru BLEU | KIWI | XCOMET |
|---|---|---|---|---|---|---|---|---|---|
| GPT-4 | 34.58 | 83.48 | 97.85 | 44.41 | 81.73 | 90.97 | 28.74 | 83.64 | 94.30 |
| Aya23-8B | 44.68 | 84.57 | 97.56 | 42.58 | 82.64 | 93.24 | 37.43 | 83.39 | 94.79 |
| LLaMA-2-13B | 13.69 | 68.33 | 90.81 | 30.00 | 74.09 | 81.06 | 0.59 | 56.78 | 84.94 |
| SFT on prefer | 30.25 | 81.38 | 88.81 | 27.94 | 80.18 | 87.94 | 27.12 | 80.29 | 88.37 |
| DPO | 29.50 | 80.61 | 90.03 | 27.33 | 78.94 | 88.23 | 26.32 | 79.33 | 87.03 |
| CPO | 30.54 | 81.53 | 90.16 | 24.87 | 80.57 | 89.85 | 25.14 | 80.22 | 89.63 |
| ICFT | 29.19 | 81.02 | 89.24 | 24.77 | 79.37 | 88.12 | 24.54 | 79.10 | 88.78 |
| ICPFT | 29.96 | 81.10 | 89.91 | 25.78 | 79.70 | 88.37 | 27.93 | 79.91 | 89.13 |
| PECPO | 31.41 | 82.41 | 90.66 | 25.89 | 80.26 | 90.23 | 26.20 | 79.51 | 90.11 |
| PEIT | 31.22 | 81.13 | 91.74 | 26.33 | 80.59 | 90.51 | 27.22 | 80.33 | 90.47 |

| Models | cs BLEU | KIWI | XCOMET | ind BLEU | KIWI | XCOMET | Avg. BLEU | KIWI | XCOMET |
|---|---|---|---|---|---|---|---|---|---|
| GPT-4 | 33.74 | 84.81 | 93.48 | 27.63 | 83.21 | 91.27 | 33.82 | 83.37 | 93.57 |
| Aya23-8B | 46.83 | 84.79 | 94.52 | 42.12 | 83.47 | 95.11 | **42.72** | **83.77** | **95.04** |
| LLaMA-2-13B | 0.87 | 61.38 | 74.26 | 12.54 | 62.35 | 75.22 | 11.53 | 64.58 | 81.25 |
| SFT on prefer | 26.22 | 79.88 | 87.62 | 25.93 | 81.46 | 89.48 | **27.49** | 80.63 | 88.44 |
| DPO | 26.25 | 79.21 | 88.68 | 25.41 | 81.77 | 89.43 | 26.96 | 79.97 | 88.67 |
| CPO | 27.13 | 80.63 | 88.73 | 27.21 | 81.35 | 90.04 | 26.97 | 80.85 | 89.68 |
| ICFT | 25.06 | 79.09 | 86.64 | 24.95 | 80.09 | 89.33 | 25.70 | 79.73 | 88.42 |
| ICPFT | 28.25 | 79.10 | 87.28 | 25.37 | 81.85 | 89.21 | 27.45 | 80.33 | 88.77 |
| PECPO | 27.88 | 80.26 | 89.27 | 23.73 | 81.38 | 89.26 | 27.02 | 80.76 | 89.90 |
| PEIT | 29.47 | 80.38 | 89.53 | 26.01 | 82.93 | 90.13 | 28.05 | **81.07** | **90.47** |

