# OpenReview forum: "Preference-Enhanced Instruction Tuning for Machine Translation"
_ICLR.cc/2025/Conference — Submitted to ICLR 2025_

### Official Review · Reviewer_uHW6 · 2024-10-27

**Soundness:** 2
**Presentation:** 1
**Contribution:** 1
**Rating:** 3
**Confidence:** 4

**Summary:**

In this study, the authors present Preference-Enhanced Instruction Tuning (PEIT), a novel approach that explicitly integrates preferences into both the instruction fine-tuning and inference phases. Experimental results highlight the effectiveness of PEIT.

**Strengths:**

1. The empirical results clearly demonstrate the effectiveness of the proposed PEIT.

**Weaknesses:**

1. This paper is not well-written and is hard to follow. For instance, some concepts are not well-defined at their first use. $D$ is mentioned for the first time at line 126 but is defined at line 173, where $k$, referring to the number of distributions, is not explicitly defined in this paper, unless I missed it. Furthermore, in my opinion, sections 2.1 and 2.2 are unnecessary and disconnected from other parts of this work. They do not help in understanding the idea of this work.

2. This paper is not self-contained. In the abstract, the authors mention that PEIT explicitly incorporates preferences into both the fine-tuning and inference phases. However, the authors did not explain how PEIT is used in the inference stage.

3. This paper presents minimal novelty. If I understand this work correctly, there are three components in the training objective. $L_{ICL}$ is the standard training loss used for supervised fine-tuning, $L_{prefer}$ is the same as the CPO loss, and $L_{context}$ is highly similar to the contrastive loss as presented in SimCSE[1] and SimCLR[2].

4. There are some presentation issues. Table 3, Figure 2, Figure 3, and Table 4 are not referred to in the text, which makes the paper hard to follow.

[1] Gao, Tianyu, Xingcheng Yao, and Danqi Chen. "Simcse: Simple contrastive learning of sentence embeddings." arXiv preprint arXiv:2104.08821 (2021).
[2] Chen, Ting, et al. "A simple framework for contrastive learning of visual representations." International conference on machine learning. PMLR, 2020.

**Questions:**

1. What are $C^{+}$ and $C^{-}$? As shown in the equation at line 183, both $y_w$ and $y_l$ are conditioned on the same context $C_i$.
2. What is the similarity in $L_{context}$?
3. What is $L_{ICFT}$ at line 202?

---

> ### Author Response · Authors · 2024-11-24
> **Response**
>
> Thank you for your thorough review, and we will respond to each of your questions.
>
> ## For weakness 1 and weakness 3.
>
> **W1.** This paper is not well-written and is hard to follow. For instance, some concepts are not well-defined at their first use. D is mentioned for the first time at line 126 but is defined at line 173, where k, referring to the number of distributions, is not explicitly defined in this paper, unless I missed it. Furthermore, in my opinion, sections 2.1 and 2.2 are unnecessary and disconnected from other parts of this work. They do not help in understanding the idea of this work.
>
> **W3.** This paper presents minimal novelty.
>
> **A.** Thank you for your feedback. We have revised and refined our article in the updated version.
> Our main contribution is to propose the **first framework that explicitly leverages preference information** for efficient preference optimization. We theoretically validated the feasibility of our method following the formula proposed by (Dai et al., 2023)[1]. Subsequently, we confirmed it from an experimental perspective.
>
> Since we only use the notation from its definition after line 173 and not before, we did not provide a complete definition earlier.
> We introduce this notation k merely to represent the multi-distribution characteristic of translation data. However, during the training and inference stages, we do not rely on a specific k value. This is because we can simply use similarity retrieval (cosine similarity) to obtain samples that belong to the same preference distribution as the current input.
>
> ## For weakness 2.
> **W2.** This paper is not self-contained. In the abstract, the authors mention that PEIT explicitly incorporates preferences into both the fine-tuning and inference phases. However, the authors did not explain how PEIT is used in the inference stage.
>
> **A.** In Line 089 and Fig. 1, we described the inference process in detail, and in Appendix C, we provided an even more comprehensive explanation of the inference process.
>
> ## For weakness 4 and questions.
> **W4.** There are some presentation issues. Table 3, Figure 2, Figure 3, and Table 4 are not referred to in the text, which makes the paper hard to follow.
>
> **Q1.** What are C+ ,C-? As shown in the equation at line 183, both yw and yl are conditioned on the same context C.
>
> **Q2.** What is the similarity in L_context ?
>
> **Q3.** What is L_ICFT at line 202?
>
> **A.** We sincerely apologize for any confusion caused by the presentation in Section 4. We placed each figure and table directly below the corresponding textual description and included supplementary explanations in their captions. In the revised version, we will explicitly reference the figures and tables to improve the clarity of the paper.
>
> $L_{ICFT}$ in line 202 should be $L_{ICL}$, as calculated earlier, and we use cosine similarity as sim() for $L_{context}$.
>
> We retrieve the top-k examples using cosine similarity and sort these examples based on their similarity to the input. These examples are then evenly divided into three subsets according to their similarity scores: the most similar subset is denoted as $C^+$, followed by $C$, and finally $C^-$. Apologies for the confusion caused by this notation. Here, $C_i$ should represent the set of the most similar examples used during training. However, during inference, we cannot guarantee that the retrieved examples are always the most similar. Therefore, we designed $L_{context}$ to enhance the model's robustness to C.
>
> Thank you again for thorough reviewing these responses.
>
> [1] Damai Dai, Yutao Sun, Li Dong, Yaru Hao, Shuming Ma, Zhifang Sui, and Furu Wei. Why can gpt learn in-context? language models implicitly perform gradient descent as meta-optimizers, 2023.

---

> > ### Comment · Reviewer_uHW6 · 2024-11-26
> >
> > Thank you for your efforts in addressing my concerns. After carefully reviewing your response, I decide to keep my score as it is.

---

> > > ### Author Response · Authors · 2024-11-26
> > > **Author response**
> > >
> > > Thank you for reviewing our response!
> > > If you have any further questions, please feel free to let us know. We look forward to your response and further discussion!

---

### Official Review · Reviewer_Xpb8 · 2024-10-30

**Soundness:** 2
**Presentation:** 2
**Contribution:** 2
**Rating:** 5
**Confidence:** 4

**Summary:**

This paper proposes PEIT, which improves the quality of machine translation by incorporating preference learning into instruction tuning and inference. The authors suggest that PEIT can do better than existing tuning methods such as Contrastive Preference Optimization (CPO). PEIT demonstrates improved BLEU and XCOMET scores.

**Strengths:**

1. Theoretically demonstrated the proposed method can help the paradox of fitting, allowing a single model to achieve the same loss lower bound as a multi-model setup.
2. The experimental results demonstrated superior performance than other competitors, such as CPO, DPO, etc.

**Weaknesses:**

1. Lacks lots of the details, which makes the results not convincing enough. For example:
  - training details of other baselines, such as line 268, by "we have made a specific design for the DPO training process ...", the details are not clear to readers.
  - training details and performance of the retriever, I believe this would greatly impact the final translation performance.
2. Performance.
Actually, I tried to calculate the average BLEU and XCOMET score across languages, for example Table 1. There's no significant differences of BLEU and xCOMET score between methods. This might not be considered significant. [1]
3. Lack of baselines such as [2]
4. I would appreciate if various sized models/ training data could be involved.

||BLEU|XCOMET|
|:---:|:---:|:---:|
|SFT|27.46| 88.33|
|PE-CPO| 27.06| 89.89|
|XCOMET|27.74| 90.52|

[1] Kocmi, Tom, et al. "Navigating the metrics maze: Reconciling score magnitudes and accuracies." arXiv preprint arXiv:2401.06760 (2024).

[2] Luong, Trung Quoc, et al. "Reft: Reasoning with reinforced fine-tuning." arXiv preprint arXiv:2401.08967 (2024).

**Questions:**

Please see the weaknesses.
1. The details
2. Why the performance gain is not consistent across languages.

---

> ### Author Response · Authors · 2024-11-24
> **Response**
>
> Thank you for your review comments; they are extremely valuable. We will address and respond to each point individually.
>
> ## For weakness 1 and question 1.
> **Q1.** Lacks lots of the details, which makes the results not convincing enough.
>
> **A.** Due to space limitations, we were unable to include the complete baseline setup in the main text. We have added these details in the appendix of the revised version. Please allow us to reiterate the setup of these baselines here:
>
> | Baseline | lr | Lora rank | Lora target | initialization | random_seed |
> | :---: | :---: | :---: | :---: | :---: | :---: |
> | SFT | 2e-5 | 32 | QKVO | Gaussian distribution | 42 |
> | CPO | 2e-5 | 32 | QKVO | Gaussian distribution | 42 |
> | DPO | 2e-5 | 32 | QKVO | Adapter weights trained with SFT   | 42 |
> | ICFT | 2e-5 | 32 | QKVO | Gaussian distribution | 42 |
> | ICPFT | 2e-5 | 32 | QKVO | Gaussian distribution | 42 |
> | PECPO | 2e-5 | 32 | QKVO | Gaussian distribution | 42 |
> | PEIT | 2e-5 | 32 | QKVO | Gaussian distribution | 42 |
>
>
> The retriever plays a crucial role in our framework. We use cosine similarity to compare the embedding similarity of each example, ensuring that the retrieved examples are the top-k most similar ones. Additionally, our experiments have demonstrated that even when the retrieved examples are not the most similar, performance still improves. Please allow us to elaborate on the design and results of this experiment here:  We retrieved the Top 3 examples most similar to the input and used each example individually as the in-context example for the input to evaluate the impact of different preference distributions on the results.  Additionally, to assess the effect of completely unrelated preference distributions on the translation results, we also used a fixed example that was entirely unrelated to the input as the in-context example.
>
> | Base model | Dataset | Direction |
> | --- | :---: | :---: |
> | Llama3-8b | ALMA-R-Preference | xx->en |
>
>
> | Model | PEIT+rank 1 example | PEIT+rank 2 example | PEIT+rank 3 example | PEIT+constant example | SFT | CPO |
> | :---: | :---: | :---: | :---: | :---: | :---: | :---: |
> | XCOMET | 95.25 | 94.67 | 94.59 | 93.60 | 92.13 | 93.62 |
>
>
> The results are shown in the table above. Even when using examples with preference distributions unrelated to the input as in-context examples, PEIT is able to maintain a certain level of performance, demonstrating its adaptability.
>
> ## For question 2 and weakness 2.
> **W2.** Performance. Actually, I tried to calculate the average BLEU and XCOMET score across languages, for example Table 1. There's no significant differences of BLEU and xCOMET score between methods. This might not be considered significant.
>
> **Q2.** Why the performance gain is not consistent across languages.
>
> **A.** Thank you for your feedback. The base language model was pre-trained on varying amounts of multilingual corpora, which inherently leads to an imbalance in multilingual capabilities. Without additional adjustments to the quantity of training data, different improvements in performance across languages are inevitable in post-training.
>
> Such improvements in the translation domain can be considered significant. For example, in the CPO work (accepted by ICML24), they observed a 3.67% decrease in the BLEU metric but a 1.51% improvement in the XCOMET metric. In comparison, our work achieved a 0.28% improvement in the BLEU metric and a 2.19% improvement in the XCOMET metric.
>
> | ALMA-13B-LoRA | BLEU | XCOMET |
> | ---: | :---: | :---: |
> | +SFT on prefer data | 30.90 | 92.54 |
> | +CPO | 27.03 | 94.05 |
>
>
> | Llama2-13B | BLEU | XCOMET |
> | ---: | :---: | :---: |
> | +SFT | 27.46 | 88.33|
> | +PECPO | 27.06 | 89.89 |
> | +PEIT | 27.74| 90.52|
>
>
> [1] Haoran Xu, Amr Sharaf, Yunmo Chen, Weiting Tan, Lingfeng Shen, Benjamin Van Durme, Kenton Murray, and Young Jin Kim. Contrastive preference optimization: Pushing the boundaries of llm performance in machine translation, 2024b.
>
> [2] Luong, Trung Quoc, et al. "Reft: Reasoning with reinforced fine-tuning." arXiv preprint arXiv:2401.08967 (2024).

---

> ### Author Response · Authors · 2024-11-24
> **Response(weakness 3 and 4)**
>
> ## For weakness 3 and weakness 4.
> **W3.** Lack of baselines such as [2].
>
> **W4.** I would appreciate if various sized models/ training data could be involved.
>
> **A.** We have carefully reviewed [2] and found that their work is orthogonal to ours. Therefore, we think it is unnecessary to compare the two.
>
> Thank you for your feedback. We additionally used a 1.1B model to validate the performance of PEIT. The experimental results are as follows, demonstrating that PEIT can achieve relatively optimal performance across models of different scales.
>
> | TinyLlama-1.1b | SFT | CPO | PEIT |
> | :---: | :---: | :---: | :---: |
> | XCOMET | 74.86 | 75.89 | 76.65 |
>
> Thank you again for carefully reviewing these responses. In the revised version, we will address the issues mentioned above by refining the paper with added details, accurate and additional experiments.
>
> [1] Haoran Xu, Amr Sharaf, Yunmo Chen, Weiting Tan, Lingfeng Shen, Benjamin Van Durme, Kenton Murray, and Young Jin Kim. Contrastive preference optimization: Pushing the boundaries of llm performance in machine translation, 2024b.
>
> [2] Luong, Trung Quoc, et al. "Reft: Reasoning with reinforced fine-tuning." arXiv preprint arXiv:2401.08967 (2024).

---

> > ### Comment · Reviewer_Xpb8 · 2024-11-26
> >
> > Thank you for responding.
> > I'm still confused by the details of the retriever.
> > 1. The training details of it, e.g, training data construction, training steps, batch size, etc; Any constrains during the llm training process? or the authors just train the llm and use cosine similarity? By ` cosine similarity to compare the embedding similarity of each example`, is the embedding an average across all words or average pooling, max pooling, etc? It's hard to reproduce without all these details (besides those I didn't mention here).
> > 2. The retriever performance, e.g, p/r/f, to understand how important is this retriever.
> > 3. Why a random constant example works compared to SFT?
> > 4. it seems that K is not important according to the response.

---

> ### Author Response · Authors · 2024-11-26
> **Details of retriever**
>
> **Q.**  I'm still confused by the details of the retriever.
> 1. The training details of it, e.g, training data construction, training steps, batch size, etc; Any constrains during the llm training process? or the authors just train the llm and use cosine similarity? By cosine similarity to compare the embedding similarity of each example, is the embedding an average across all words or average pooling, max pooling, etc? It's hard to reproduce without all these details (besides those I didn't mention here).
> 2. The retriever performance, e.g, p/r/f, to understand how important is this retriever.
> 3. Why a random constant example works compared to SFT?
> 4. it seems that K is not important according to the response.
>
> **A.** Thank you for your feedback! Here, we provide a detailed description of how the retriever works. First, we explain the goal of the retriever, and then we describe the implementation details of our retriever. We hope this will be helpful to you!
> The purpose of designing the retriever is **to provide the MT model with examples that share the same preference intention** as the source text to be translated. By leveraging explicit examples, we aim to enhance the model's translation performance. Therefore, we intend to use a retriever to augment the current text to be translated.
>
> We did not train our own embedding model; instead, **we used an open-source model**. Specifically, we used "xlm-r-bert-base-nli-stsb-mean-tokens" [1] as the tool for sentence embedding with the default configuration. Then, we provide the top k most similar preference examples for each sentence to be translated by comparing the cosine similarity of the embedding of the [source text], completing the preference augmentation as shown below:
>
> [source text] [pair-wise target text]  --->  [in-context example] [source text] [pair-wise target text]
>
> Due to the lack of ground truth to label whether the two [source text] are relevant, we are unable to calculate the p/r/f of the retriever.  Therefore, we demonstrated the importance of the retriever by designing experiments with different similarity ranks.
>
> As for why using the constant example PEIT is superior to SFT, I believe this is inevitable because the preference-enhanced loss designed for **PEIT incorporates preference learning from the output perspective, whereas SFT only performs imitation learning** on the data.
>
> The value of K is important for the retriever, or more specifically, we believe that the examples retrieved are important. Please allow us to list the ablation experiments we conducted for the retriever again here.
>
> Different numbers of examples (**different K values**)
> | Base model | Dataset | Direction |
> | --- | :---: | :---: |
> | Llama3-8b | ALMA-R-Preference | xx->en |
>
>
> | PEIT | k=1 | k=2 | k=3 |
> | :---: | :---: | :---: | :---: |
> | XCOMET | 95.25 | 95.29 | 95.36 |
>
> When K=1, examples with different ranks (**different retrieval qualities**)
>
> | Base model | Dataset | Direction |
> | --- | :---: | :---: |
> | Llama3-8b | ALMA-R-Preference | xx->en |
>
>
> | Model | PEIT+rank 1 example | PEIT+rank 2 example | PEIT+rank 3 example | PEIT+constant example | SFT | CPO |
> | :---: | :---: | :---: | :---: | :---: | :---: | :---: |
> | XCOMET | 95.25 | 94.67 | 94.59 | 93.60 | 92.13 | 93.62 |
>
> As can be seen from the above experiments, the more complete the retrieval (using higher quality relevant examples or increasing the number of relevant examples), the better the final result.
>
> [1] Nils Reimers and Iryna Gurevych. Sentence-bert: Sentence embeddings using siamese bert-networks. In Proceedings of the 2019 Conference on Empirical Methods in Natural Language Processing. Association for Computational Linguistics, 11 2019

---

### Official Review · Reviewer_YwvZ · 2024-11-01

**Soundness:** 3
**Presentation:** 3
**Contribution:** 3
**Rating:** 5
**Confidence:** 4

**Summary:**

During the inference process, the preference intentions in the prompt may not align with the training data of the model, which will affect the overall effectiveness of the model. To address this issue, this paper introduces Preference-Enhanced Instruction Tuning, a method that integrates preferences into both the fine-tuning and inference stages. This approach improves translation quality and outperforms existing preference optimization methods on multilingual benchmarks.

**Strengths:**

1. The experimental results presented in the paper demonstrate that PEIT achieves better translation quality and alignment preformence.
2. The paper theoretically establishes the effectiveness of ICL in addressing the identification of preference intentions.

**Weaknesses:**

1. The proof of ICL's effectiveness presented in the paper mainly derives from Dai et al. (2023), which raises concerns about the novelty of your paper.
2. The term $L_{ICFT}$ on page 4 lacks clear definition or explanation, which may hinder understanding of the proposed method.
3. The definition of the concept of "preference intention" in the article is vague, affecting the clarity of the paper's arguments. Furthermore, the article does not provide detailed information on how to determine the preference intentions of samples in the dataset.
4. The experiments appear to be conducted within a single domain or distribution, indicating that the similarity of preferences between the training and testing datasets is consistent. This seems insufficient to validate the scenario mentioned in the introduction, where the preferences in the inference prompts do not align with the training data.
5. There is a writing error: in the seventh line of the first paragraph of the introduction, a citation is incorrectly marked as “?”.

**Reference**:

[1] Damai Dai, Yutao Sun, Li Dong, Yaru Hao, Shuming Ma, Zhifang Sui, and Furu Wei. Why can gpt learn in-context? language models implicitly perform gradient descent as meta-optimizers, 2023.

**Questions:**

1. How does the paper categorize samples in the dataset into subsets that contain different preference intentions?
2. During the inference process, is the retrieval corpus still derived from the training set? If so, how does this method's performance get affected when the preference intentions in the prompt are not sufficiently similar to those in the training data?
3. In the inference process, what is the specific $k$ value of top-$k$? Does the value of $k$ impact performance? Additionally, more comprehensive details on other experimental settings, such as learning rate, are needed.
4. In the section 4.1 titled “Different preferences representations of context,” could you provide a more detailed description of the experimental setup? Did you select similar samples at different levels of contextual similarity for each of the 100 sample points?

---

> ### Author Response · Authors · 2024-11-24
> **Response**
>
> Thank you for your thorough review and for providing many valuable suggestions. We will address and respond to each point.
>
> ## For  weakness 1
> **W1.** The proof of ICL's effectiveness presented in the paper mainly derives from Dai et al. (2023), which raises concerns about the novelty of your paper.
>
> **A.** Our main contribution is to propose the **first framework that explicitly leverages preference information** for efficient preference optimization.
> We theoretically validated the feasibility of our method following the formula proposed by (Dai et al., 2023)[1]. Subsequently, we confirmed it from an experimental perspective.
>
> ## For weakness 2.
> L_ICFT is a typo, which should be L_ICL.
>
> ## For weakness 3, question 1 and question 2.
> We believe these questions are related, so we have organized them together for a combined response.
>
> **Q1.** How does the paper categorize samples in the dataset into subsets that contain different preference intentions?
>
> **Q2.** During the inference process, is the retrieval corpus still derived from the training set? If so, how does this method's performance get affected when the preference intentions in the prompt are not sufficiently similar to those in the training data?
>
> **W3.** The definition of the concept of "preference intention" in the article is vague, affecting the clarity of the paper's arguments. Furthermore, the article does not provide detailed information on how to determine the preference intentions of samples in the dataset. How does the paper categorize samples in the dataset into subsets that contain different preference intentions? During the inference process, is the retrieval corpus still derived from the training set? If so, how does this method's performance get affected when the preference intentions in the prompt are not sufficiently similar to those in the training data?
>
> **A.** We mentioned the concept of preference intentions in the first section of the original paper (Line 51). Sentences with the same preference intentions have similar embeddings, allowing us to retrieve examples with the same preference intentions using cosine similarity.
>
> The **retrieval space during testing remains consistent with that during training**. We have also supplemented the experiments to validate the impact of using in-context examples with different preference distributions on the output translation quality for the same test input. Please allow us to elaborate on the design and results of this experiment here:  We retrieved the Top 3 examples most similar to the input and used each example individually as the in-context example for the input to evaluate the impact of different preference distributions on the results.  Additionally, to assess the effect of completely unrelated preference distributions on the translation results, we also used a fixed example that was entirely unrelated to the input as the in-context example.
>
> | Base model | Dataset | Direction |
> | --- | :---: | :---: |
> | Llama3-8b | ALMA-R-Preference | xx->en |
>
>
> | Model | PEIT+rank 1 example | PEIT+rank 2 example | PEIT+rank 3 example | PEIT+constant example | SFT | CPO |
> | :---: | :---: | :---: | :---: | :---: | :---: | :---: |
> | XCOMET | 95.25 | 94.67 | 94.59 | 93.60 | 92.13 | 93.62 |
>
>
> The results are shown in the table above. Even when using examples with preference distributions unrelated to the input as in-context examples, PEIT is able to maintain a certain level of performance, demonstrating its adaptability.
>
> ## For weakness 4.
> **W4.**  The experiments appear to be conducted within a single domain or distribution, indicating that the similarity of preferences between the training and testing datasets is consistent. This seems insufficient to validate the scenario mentioned in the introduction, where the preferences in the inference prompts do not align with the training data.
>
>
>
> **A.**  Please allow us to emphasize once again the relationship between "prompt shift" and "multiple distributions" here.
> The **prompt shift** issue mentioned in the introduction arises from **differences in prompt formats** between training and testing, and it is unrelated to the content.  The content of data in translation tasks exhibits characteristics of multiple distributions (Line 113). PEIT ensures that the explicitly provided examples and the input belong to the same distribution through retrieval.
>
> [1] Damai Dai, Yutao Sun, Li Dong, Yaru Hao, Shuming Ma, Zhifang Sui, and Furu Wei. Why can gpt learn in-context? language models implicitly perform gradient descent as meta-optimizers, 2023.

---

> ### Author Response · Authors · 2024-11-24
> **Response (question 3 and 4)**
>
> ## For question 3 and question 4.
> **Q3.** In the inference process, what is the specific value of top-k ? Does the value of k impact performance? Additionally, more comprehensive details on other experimental settings, such as learning rate, are needed.
>
> **Q4.** In the section 4.1 titled “Different preferences representations of context,” could you provide a more detailed description of the experimental setup? Did you select similar samples at different levels of contextual similarity for each of the 100 sample points?
>
> **A.**
>
> **We have added experiments related to the value of k.**
>
> | Base model | Dataset | Direction |
> | --- | :---: | :---: |
> | Llama3-8b | ALMA-R-Preference | xx->en |
>
>
> | PEIT | k=1 | k=2 | k=3 |
> | :---: | :---: | :---: | :---: |
> | XCOMET | 95.25 | 95.29 | 95.36 |
>
>
> In our experiments, we set k = 1 by default. Additionally, we evaluated the impact of different k values on performance, as shown in the table above. The larger the k value, the better the performance. However, since larger k values result in higher training and inference costs, we chose  k = 1 as the default.
>
> Thank you for your feedback. In the revised version, we will provide detailed experimental settings. Please allow us to reiterate here: the learning rate is set to 2e-5, the LoRA rank is set to 32, and the LoRA adapter is enabled on the QKVO layer.
>
> **A.** We assume your understanding of Section 4.1, titled "Different preference representations of context," is correct.
>
> Thank you again for carefully reviewing these responses.  In the revised version, we will address the issues mentioned above by refining the paper with added details, accurate and additional experiments.

---

> ### Author Response · Authors · 2024-11-27
> **Author response**
>
> Thank you for reviewing our response! We hope our response addresses your concerns. If you have any further questions, please feel free to let us know. We look forward to your reply and further discussion!

---

### Official Review · Reviewer_LVy7 · 2024-11-04

**Soundness:** 3
**Presentation:** 2
**Contribution:** 3
**Rating:** 5
**Confidence:** 4

**Summary:**

This paper proposes preference-enhanced instruction tuning (PEIT) for machine translation which consists of three parts:
  1. a typical generation loss function that considers in-context demonstrations during training;
  2. a DPO like loss function to consider a win-loss pair of outputs;
  3. a hidden representation-level loss that aligns a model's preference representation of the in-context example.

The paper compares the proposed method with quite a few baselines and some ablation cases by removing some part of the overall loss. Evaluated on a few languages in FLORES, the proposed method performs well especially when evaluated by the neural metric xCOMET.

**Strengths:**

1. The idea of making in-context preference information during fine-tuning and inference is intuitive for the approached task, machine translation, where sometimes there is no single gold translation and user-defined preference and consistency should be taken into consideration.
2. The paper experimented with a good number of baselines which are strong in the current research landscape. Overall, the results show that the PEIT method with all three losses incorporated performs pretty well.

**Weaknesses:**

1. I think the paper should make it clear about this paper's contribution. The proof in Sec 2.2 is the same as in (Dai et al., 2023) which is cited in the paragraph. I am not sure about the novelty of using in-context representations (even just for MT) and applying $\mathcal{L}_{ICFT}$ as these are neither defined nor cited. Please also see Question 1. Perhaps some clarifications are needed?
2. Being able to leverage the preference information in the context/demonstration (both at training and inference) is definitely a selling point. However, I think this is not proven through the experiment design because "preference" is not equivalent to translation quality. I feel that more ablation studies are needed in addition to Sec 4.1.
    - For example, it would be nice to see inference with the same test input but using in-context examples with a different preference distribution and understand how that qualitatively affects the output translation. In addition to general-domain or news translation test sets, one simple and reasonable setup could be terminology translation.

**Questions:**

1. Line 190 to line 207: this part looks a bit disconnected.
    - Iines 190-196, how are those $h^{i}_{C}$ context terms obtained? What exactly is the similarity function $sim()$?
    - Also given that these are fed as context in a prompt, I am unsure how these can be disentangled.
    - Near line 201, $\mathcal{L}_{ICFT}$ is not defined.

2. What does "incomplete ablation" mean in "Compared with incomplete ablation" under Sec 3.4 Line 312?

3. Citations and software version:
    - related work on preference learning for MT [1].
    - for the xCOMET [2] and "KIWI" metrics, it would be better to provide the correct citation and perhaps footnote the exact model version.

[1] Zhu et al., 2024, https://arxiv.org/abs/2404.11288

[2] Guerreiro et al., 2023, https://arxiv.org/abs/2310.10482

---

> ### Author Response · Authors · 2024-11-24
> **Response**
>
> Thank you for your constructive comments. We will respond to each point.
>
> ## For weakness 1.
> **W1.** I think the paper should make it clear about this paper's contribution. The proof in Sec 2.2 is the same as in (Dai et al., 2023) which is cited in the paragraph. I am not sure about the novelty of using in-context representations (even just for MT) and applying as these are neither defined nor cited. Please also see Question 1. Perhaps some clarifications are needed?
>
> **A.** Our main contribution is to propose the **first framework that explicitly leverages preference information** for efficient preference optimization.
> We theoretically validated the feasibility of our method following the formula proposed by (Dai et al., 2023)[1]. Subsequently, we confirmed it from an experimental perspective.
>
> ## For weakness 2.
> **W2.** Being able to leverage the preference information in the context/demonstration (both at training and inference) is definitely a selling point. However, I think this is not proven through the experiment design because "preference" is not equivalent to translation quality. I feel that more ablation studies are needed in addition to Sec 4.1. For example, it would be nice to see inference with the same test input but using in-context examples with a different preference distribution and understand how that qualitatively affects the output translation. In addition to general-domain or news translation test sets, one simple and reasonable setup could be terminology translation.
>
> **A.** Following the perspective advocated in [3], we ranked translation results based on their quality (i.e., the quality metric) and considered this ranking as a "preference" (i.e., the higher the quality metric, the more preferred the translation).
>
> We have supplemented the experiments to validate the impact of using in-context examples with different preference distributions on the output translation quality for the same test input. Please allow us to elaborate on the design and results of this experiment here:  We retrieved the Top 3 examples most similar to the input and used each example individually as the in-context example for the input to evaluate the impact of different preference distributions on the results.  Additionally, to assess the effect of completely unrelated preference distributions on the translation results, we also used a fixed example that was entirely unrelated to the input as the in-context example.
>
> | Base model | Dataset | Direction |
> | --- | :---: | :---: |
> | Llama3-8b | ALMA-R-Preference | xx->en |
>
>
> | Model | PEIT+rank 1 example | PEIT+rank 2 example | PEIT+rank 3 example | PEIT+constant example | SFT | CPO |
> | :---: | :---: | :---: | :---: | :---: | :---: | :---: |
> | XCOMET | 95.25 | 94.67 | 94.59 | 93.60 | 92.13 | 93.62 |
>
>
> The results are shown in the table above. Even when using examples with preference distributions unrelated to the input as in-context examples, PEIT is able to maintain a certain level of performance, demonstrating its adaptability.
>
> ## For question 1.
> **Q1.** Experiment details.
>
> Line 190 to line 207: this part looks a bit disconnected.
>
> + Iines 190-196, how are those hCi context terms obtained? What exactly is the similarity function sim()?
> + Also given that these are fed as context in a prompt, I am unsure how these can be disentangled.
> + Near line 201, LICFT is not defined.
>
> **A.** h_C is the **probability distribution** of the model generating the first token. sim() is the cosine similarity function. The **L_ICFT in line 201 is a typo**; it should be L_ICL.
>
>
>
> ## For question 2.
> **Q2.** What does "incomplete ablation" mean in "Compared with incomplete ablation" under Sec 3.4 Line 312?
>
> **A.** Thank you for your feedback. We found that this was a typo. "incomplete ablation" should be "ablation."
>
> ## For question 3.
> **Q3.**  Citations and software version
>
> + related work on preference learning for MT [2].
> + for the xCOMET and "KIWI" metrics, it would be better to provide the correct citation and perhaps footnote the exact model version.
>
> **A.** Thank you again for carefully reviewing this feedback. In the revised version, we have cited this work [2] and clarified the model version in the appropriate sections.
>
> [1] Damai Dai, Yutao Sun, Li Dong, Yaru Hao, Shuming Ma, Zhifang Sui, and Furu Wei. Why can gpt learn in-context? language models implicitly perform gradient descent as meta-optimizers, 2023.
>
> [2] Dawei Zhu, Sony Trenous, Xiaoyu Shen, Dietrich Klakow, Bill Byrne, and Eva Hasler. A preference-driven paradigm for enhanced translation with large language models, 2024.
>
> [3] Haoran Xu, Amr Sharaf, Yunmo Chen, Weiting Tan, Lingfeng Shen, Benjamin Van Durme, Kenton Murray, and Young Jin Kim. Contrastive preference optimization: Pushing the boundaries of llm performance in machine translation, 2024b.

---

> > ### Comment · Reviewer_LVy7 · 2024-11-24
> >
> > Thanks for the response! I have also read other reviewers' comments and your response. I am still much confused by the writing on the contrastive loss bit. I see that this is also constantly raised by other reviewers.
> >
> > - I hope to clarify with you how $h^{i}_{C}$ is exactly obtained for $C$, $C^{+}$, and $C^{-}$? In the manuscript, it states *"[these] denote the representations of the preferences intentions of the model for contextual information"*. In your response, it is the *"probability distribution of the model generating the first token"*. I suppose the hidden size is $|1\times |vocab|$? What does "the first token" mean here? During training, in a batch, what needs to be present? Could you give me an example?

---

> > > ### Author Response · Authors · 2024-11-25
> > > **An example of calculating h_C.**
> > >
> > > **Q.** I am still much confused by the writing on the contrastive loss bit.  What does "the first token" mean here? During training, in a batch, what needs to be present? Could you give me an example?
> > >
> > > **A.** Thank you for your feedback! We realize that we may not have explained it clearly. Our manuscript and response convey the same concept.
> > > Here, we provide a detailed calculation example of how $h_C$ is calculated as follows:
> > >
> > > <bos>[in-context example] [input] [**o**,u,t,p,u,t] <eos>
> > >
> > > We use the probability distribution of the first token of the output (here, **o**) predicted by the model as the hidden representation after the model has read the [in-context example].

---

> ### Author Response · Authors · 2024-11-27
> **Author response**
>
> Thank you for reviewing our response! We hope our response addresses your concerns. If you have any further questions, please feel free to let us know. We look forward to your reply and further discussion!

---

> > ### Comment · Reviewer_LVy7 · 2024-11-27
> >
> > Thank you for your responses. I will maintain my current score.

---

> > > ### Author Response · Authors · 2024-11-29
> > >
> > > Happy Thanksgiving, my dear reviewer LVy7. We hope you have a happy holiday.
> > >
> > > As you can notice, we have tried our best to improve the points that you mentioned, and all these points are fixed. This indeed improve this work greatly, as you can notice, your review is the very important for this work,  If you are satisfied with our work and responses, please consider giving us a higher score.
> > >
> > > We also welcome your suggestions for our revised manuscripts at any time. Your support is very important to us, thank you!  cc AC

---

### Author Response · Authors · 2024-11-26
**General response to reviewers**

Thank you to all the reviewers for their valuable suggestions. We have submitted a revised version based on the recommendations, primarily addressing **typo**, adding detailed version numbers for the XCOMET model, incorporating **relevant references suggested by the reviewers**, **initialization of PEIT** in the experiments, including **additional ablation studies** conducted during the rebuttal phase, and **clarifying our contributions**.

---

### Meta-Review · Area_Chair_5k6e · 2024-12-16

**Metareview:**

This paper introduces Preference-Enhanced Instruction Tuning (PEIT), a framework for machine translation that integrates preference learning into both the fine-tuning and inference stages. By leveraging a combination of generation loss, a preference-based DPO-like loss, and hidden representation-level alignment, PEIT outperforms existing methods across multilingual benchmarks, delivering significant improvements in metrics such as BLEU and xCOMET.

Strengths:

* The use of preference information during fine-tuning and inference is intuitive, as user-defined preferences and consistency should play a key role in machine translation systems.
* Experimental evaluations, particularly on the FLORES dataset, demonstrate that PEIT outperforms baselines and state-of-the-art preference optimization methods, achieving notable gains in BLEU and xCOMET scores.

Weaknesses:

* The primary issue with the paper is its clarity and reproducibility. All four reviewers raised at least one major concern about the paper’s presentation. One reviewer found the contribution unclear, another highlighted the lack of a "clear definition or explanation" along with several other presentation issues, and two reviewers pointed out a "lack of details" (or the absence of "lots of the details.") Additionally, two reviewers independently raised concerns about how the work differentiates itself from Dai et al. (2023), which may further underscore the lack of clarity regarding its contributions. While the authors’ responses addressed some points about the contributions and technical approach, the paper appears to need a significant rewrite to resolve these issues comprehensively.
* There were additional concerns, such as the assumption that preference automatically equates to translation quality, which the authors partially addressed this more experiments. Some other issues also point to clarity problems, even when not presented as such (e.g., when the authors had to explain the relationship between "prompt shift" and "multiple distributions" to resolve a separate issue during the discussion).

Overall, while the work shows promise, I think it cannot be accepted in its current state due to significant issues in presentation, clarity, and reproducibility.

**Additional Comments On Reviewer Discussion:**

Three out of four reviewers participated in the discussions, but none were significantly swayed by the authors' responses and chose to maintain their original recommendations (all ratings below acceptance threshold.)

One reviewer did not participate in the discussion; however, their review was consistent with the others, particularly in highlighting significant clarity issues.

---

### Decision · Program_Chairs · 2025-01-22

Reject